# New Insight on Insulinoma Treatment in a Pet Rat—A Case Report

**DOI:** 10.3390/ani12202783

**Published:** 2022-10-15

**Authors:** Agata Godlewska, Karolina Barszcz, Aleksandra Orzechowska, Aleksandra Małek-Sanigórska

**Affiliations:** 1Department of Morphological Sciences, Institute of Veterinary Medicine, Warsaw University of Life Sciences SGGW, 02-787 Warsaw, Poland; 2Veterinary Clinic Dr. Wet in Warsaw, 02-796 Warsaw, Poland; 3EDINA Veterinary Clinic PulsVet in Warsaw, 02-801 Warsaw, Poland

**Keywords:** insulinoma, pancreatic tumor, pancreatic islet cell adenoma, hypoglycemia, paraparesis, posterior weakness, rat

## Abstract

**Simple Summary:**

Insulinomas are tumors of the pancreas that cause hypoglycemia. They have high prevalence in ferrets, whereas in rats are a much rare finding. The available literature on spontaneous insulinomas in rats is currently scarce. The patient from our case report was presented with no obvious hypoglycemic signs, only progressive weakness of the hind limbs was noted. However, a blood exam revealed low blood glucose and a mass in the pancreas was found on the abdominal ultrasound. The patient responded well to treatment with oral dexamethasone and survived nearly 4 months in good general condition. In geriatric rats peripheral neuropathy and pituitary tumors are very common, leading to mobility disfunction. That is why insulinoma should always be considered in cases of neurological deficits in these patients.

**Abstract:**

Insulinomas are insulin-producing tumors of pancreatic beta cells that cause hypoglycemia. They are extremely common in ferrets but have also been reported in guinea pigs and rats. This is a case report of an older rat with spontaneous insulinoma, which was confirmed by histopathology. The patient was presented at a regular check-up due to a chronic respiratory disease. The owner noticed progressive weakness of the hind limbs, which is quite commonly seen in older rats. A blood exam revealed hypoglycemia, which could have been associated with paraparesis. The patient responded to treatment with oral dexamethasone and was regularly monitored. It survived nearly 4 months in good general condition. The rat finally died most probably due to severe hypoglycemia caused by progression of the pancreatic tumor. This is the first report of a pet rat with insulinoma that was successfully treated with glucocorticoids.

## 1. Introduction

Insulinomas have high prevalence in ferrets, usually presenting with episodic dullness, irritability, depression, ataxia, muscle fasciculations, seizures and collapse. Sometimes, the leading symptom is weakness and paresis of the hind limbs [1,2,3,4]. Insulinomas have also been reported in guinea pigs, though they are relatively less common in this species. Generalized weakness, depression, ataxia, seizures and head tilt are the main clinical findings in a hypoglycemic guinea pig [4,5,6,7].

Spontaneous insulinomas in pet rats have been reported by Adissu et al. [8] and Robertson et al. [9]. In the first article, no treatment was attempted. The second one describes clinical improvement shortly after injection of dexamethasone but unsatisfactory long-term response to oral prednisone, even at high dosages. 

This is the first report of a pet rat with insulinoma that was successfully treated with dexamethasone given per os. Blood glucose remained within the reference range for most of the time, and it lived almost 4 months in general good condition from the time the diagnosis was made.

## 2. Case Report

A 28-month-old spayed female pet rat (*Rattus norvegicus*) was presented at a regular check-up due to a chronic respiratory disease that was first recognised at the age of 1 year. The patient had been on doxycycline (Unidox Solutab, Astellas Pharma Inc., Tokyo, Japany) and theophylline (Theospirex, Biofarm, Poznań, Poland) for the last 3 months and had budesonide (Nebbud, Teva, Tel Aviv, Israel) treatment with a nebulizer, but only when needed. Recently, the owner noted ambulation difficulties concerning only the hind legs.

On presentation, the rat was bright and alert, the body weight was 321 g, and there had been a slow weight loss observed over the last few weeks. The body condition score (BCS) was evaluated as 3/5. The patient had mild dyspnea, slightly worsening upon stress, but approximately the same as in over the last months. The respiratory sounds over the lungs were found to be abnormal. The rat showed signs of paraparesis. Muscle loss and reduced muscle tone in the hind limbs were noticed with impaired proprioception and decreased flexor reflex; however, nociception was present. A full blood exam was performed. Whole blood was taken from the lateral tail vein in an amount of 0.8 mL under isoflurane (Isotek, Laboratorios Karizoo, Barcelona, Spain) anesthesia—5% for induction in a chamber and 2% afterwards for maintenance via facemask (Figure 1). All parameters, besides glucose at 20.88 mg/dL, appeared to be within the range limits (Table 1).

Because of the good overall health and condition of the patient, the main suspicion for hypoglycemia in this case was insulinoma. However, differential diagnosis for hypoglycemia should also include hepatic failure, sepsis, glucose-utilizing tumor, hypoadrenocorticism, intoxication, side effects of some drugs, glycogen storage disease and prolonged anorexia. Abdominal ultrasound was performed, and it revealed a circular, hypoechoic mass in the pancreatic topography, measuring 3.6 × 2.8 mm, which could be an enlarged lymph node or insulinoma (Figure 2).

Glucocorticoid treatment with dexamethasone (Pabi-dexamethason, Adamed Pharma S.A., Pieńków, Poland) was initiated. The starting dose was 0.05 mg/kg, i.e., 1/32 of a tablet BID (bis in die—twice daily). Doxycycline, theophylline and nebulizations with budesonide were continued.

The owner was advised to modify the diet of her rat by withdrawing sugary treats and paying attention to the regularity of meals. In addition, it was recommended to measure glucose levels at home using a handheld glucometer from a blood drop taken from the tip of the tail. Performing full glucose curves appeared to be impossible due to the stress that the rat was undergoing during this procedure. Once every few days, the owner measured blood glucose 3 times a day. Based on those results the final dosage of dexamethasone (1/36 of a tablet, BID) was established. The patient remained in a good condition; her activity level was stable, while her body weight decreased. Gentle massaging of the muscles of the hind legs approximately 3 times a day for 10–15 min was recommended.

After 5 weeks of taking dexamethasone, at a dosage of 1/36 of a tablet BID for the last 3 weeks, the rat was behaving normally, and nebulizations were withheld by the owner. The body weight was 296 g (BCS was still 3/5), the patient was slightly dehydrated, and progression of muscle atrophy was noted. Dyspnea did not change, and the rat was breathing quietly. The blood exam was repeated, again under isoflurane anesthesia (Table 1). The glucose level was within the range limit at 94.5 mg/dL, while liver and kidney parameters were slightly elevated. Liver parameters might have been raised due to the glucocorticoid treatment. Raised hematocrit and total protein suggested slight dehydration. Hepatoprotectives were recommended, and the owner was asked to increase the caloric intake and fluid level in the rat’s diet by giving baby food containing meat.

Another visit took place after a month because of loud breathing and worsening of dyspnea. The body weight was 298 g (BCS 3/5), and dyspnea was exacerbated. Upon auscultation, respiratory sounds over the lungs and upper airways were abnormal. In addition, a murmur over the heart could be heard. An echocardiographic exam showed minor changes that could have been associated with the age of the patient. A chest X-ray was done, but the images were quite similar to those made 4 months prior. It was suspected that the exaggerated respiratory distress was related to the upper respiratory tract. The glucose level as measured by a handheld glucometer was 78 mg/dL. Budesonid (Nebbud, Teva, Tel Aviv, Israel), as well as fenoterol and ipratropium in a combined preparation (Berodual, Boehringer ingelheim, Ingelheim am Rhein, Germany), was recommended via nebulization twice daily. All the other drugs were continued. Symptoms resolved in the meantime.

The next check-up after another month of treatment showed further progression of muscle atrophy and resulting pododermatitis. There were no changes in the patient’s activity level, and the body weight had increased to 310 g (BCS 3/5). The rat was breathing quietly, and the dyspnea was typical for the patient. The dexamethasone dosage was increased to 1/32 of a tablet BID, because the glucose level measured 2 h after a meal was under the lower end of the range (48 mg/dL). The glucose level measured at home was still lower than 60 mg/dL, and the dosage was increased again to 1/28 of a tablet BID.

Shortly after that, nearly 4 months after starting the treatment, at the age of 2 years and 8 months, the rat’s condition suddenly deteriorated. According to the owner’s report, the rat was presenting generalized muscular weakness, was anxious and was breathing fast. The owner did not recognize those symptoms as related to hypoglycemia, so she did not initiate hypoglycemic emergency treatment. When the rat started to have seizures, the owner was already on her way to the clinic, but the rat died before getting there. The necropsy revealed a mass in the pancreas of approximately 5 mm in diameter and a local atelectasis (Figure 3). The pancreas and a fragment of a sciatic nerve were sent for a histopathologic examination. The diagnosis was islet cell adenoma, most likely derived from pancreatic islet beta cells (insulinoma), and a slight degree of inflammation of the sciatic nerve. The etiology of the diagnosed inflammation was difficult to establish in a histopathological examination.

## 3. Discussion

There are numerous reports of induced insulinomas in laboratory rats [12,13,14,15,16]. On the other hand, spontaneous islet cell tumors in laboratory rats are a rare finding [17]. Studies conducted by Bomhard et al. [18] and Walsh et al. [19] have shown similar results, with frequencies of 3.4–3.8% and 1.6% in males and females, respectively. The incidence is two times higher in males than in females. Spontaneous pancreatic islet cell neoplasms in the rat are mainly of beta cell origin [20]. In dogs, most insulinomas are carcinomas, while in rats they are predominantly adenomas [17,20].

Retrospective studies have shown that pancreatic islet cell adenomas in rats are related to age [17,21]. According to our observations, they were all diagnosed in patients over 2 years of age. In older rats, spontaneous proliferation of pancreatic endocrine tissue and fibrosis are observed histopathologically [22]. As far back as nearly 40 years ago, Turnbull et al. found evidence that increased body weight was correlated with higher risk of islet cell tumors in male rats [23]. Keenan et al. studied the effects of diet and dietary restriction on age-related proliferative and degenerative lesions. They concluded that caloric restriction improved survival and lowered the incidence and severity of the mentioned changes, as well as of many other diseases such as pituitary tumors, chronic renal disease and cardiomyopathy [24]. According to Molon et. al. the incidence of insulinomas was higher in rats fed ad libitum [25]. Overfeeding resulted in early degenerative and proliferative changes in the pancreatic islet cells, leading to fibrosis, focal hyperplasia and adenomas in rats at the age of 2 years [25].

The diet of a patient with suspected insulinoma should be low in simple sugars, and sugary treats should be withdrawn. The owners should also feed the rat a few regular small meals each day [2]. Functional pancreatic islet cell tumors produce insulin in excess, leading to hypoglycemia. Clinical signs of low blood glucose in rats may include lethargy/underactivity, piloerection, reduced body temperature, hunched posture, flat posture, collapse, irregular breathing and convulsions [14,26]. Research conducted on laboratory rats showed that prolonged hypoglycemia may cause peripheral neuropathy and paraparesis by leading to myelinated axonal damage, whereas hyperinsulinemia was associated with increased densities of small axons and endoneurial microvessels with microangiopathic changes [15,16,27,28].

In contrast to two previous reports on insulinomas in rats, which were confirmed via histopathology along with peripheral polyneuropathy, our case study revealed inflammation of the sciatic nerve of unknown etiology with no degenerative changes. According to our observations, even in severe hypoglycemic states, i.e., 20–30 mg/dL, no other clinical symptoms than paraparesis are usually noticed.

In geriatric rats, radiculoneuropathy and pituitary tumors are very common [29], leading to neurologic disorders and mobility dysfunction. That is why insulinoma should be considered as a differential diagnosis in every case of spontaneous paraparesis in a pet rat.

There are some other causes of hypoglycemia that should be taken into account in differential diagnosis. In small animals, hypoglycemia may be related to sepsis, hepatic disease, glucose-utilizing tumors, hypoadrenocorticism, intoxication, side effects of some drugs, glycogen storage disease and malnutrition [4,30]. In our experience, pancreatic tumors are the most common reason for rats to be hypoglycemic. It is important to remember that prolonged storage of whole blood promotes glucose consumption by red blood cells. Nevertheless, even if blood is stored overnight under refrigerated conditions without being centrifuged, glucose results never drop below 60 mg/dL in a healthy rat.

Suspicion of insulinoma in dogs is based on clinical signs, hypoglycemia (blood glucose concentration less than 60 mg/dL) and concurrent hyperinsulinemia. A final diagnosis can only be made through histopathologic examination of a pancreatic mass along with immunohistochemistry [4,30]. In a pet rat, we base our suspicion of insulinoma on low blood glucose results and ultrasound findings. We highly recommend performing abdominal ultrasound in every case of a suspected pancreatic tumor. An experienced ultrasonographer is usually able to find a mass within the pancreas. A typical insulinoma is round with no visible capsule, hypoechoic and homogenous, most commonly measuring 3–5 mm in diameter. Its ultrasound image is quite similar to those seen in dogs and ferrets.

In our opinion, older rats usually have concurrent diseases that make them high-risk patients for major surgery, and that is why we normally do not recommend surgical approach and opt for pharmacotherapy instead.

In other species, hypoglycemia associated with an insulinoma can be treated with glucocorticoids, diazoxide and synthetic somatostatin derivatives. Glucocorticoids increase gluconeogenesis, decrease glucose uptake by tissues and stimulate glucagon secretion, leading to higher blood glucose levels [4,30]. They can be given parenterally or orally in a stable patient. In ferrets, the most commonly used glucocorticoid is prednisone. Doses of 0.25 to 2 mg/kg BID have been used [1,3,10]. We have observed that hypoglycemic rats do not respond to treatment with prednisone at a dosage of 0.5–1 mg/kg BID. Higher doses cause them to lose weight and contribute to an increase in liver parameters. Robertson et al. presented a case report of a rat with insulinoma treated with one dose of dexamethasone given intramuscularly. The animal was discharged on oral prednisone. There was an increase in glucose level and clinical improvement at the beginning, but the patient deteriorated after 2 weeks of treatment, even though the dosage of prednisone was increased to 1.5 mg/kg BID [9]. In our opinion, this could be due to the fact that dexamethasone was discontinued. That is why we have been prescribing oral dexamethasone to hypoglycemic rats, as it works at a dosage of only 0.05 mg/kg BID. The 0.5 mg dexamethasone tablet for a typical pet rat (b. wt 300–500 g) needs to be divided into 20–30 portions. Regardless of this impediment, owners are usually capable of doing it by themselves. There is always an option to ask a pharmacist to divide the drug precisely.

The glucocorticoid treatment becomes less effective with time due to continued growth of the tumor, which leads to the release of more insulin. It requires a gradual increase in the glucocorticoid dose [2]. However, it should be remembered that long-term treatment with oral glucocorticoids, especially in high doses, may cause potentially serious adverse effects such as behavioral changes, adrenal insufficiency, cardiovascular disease, hypertension, hyperglycemia and diabetes, osteoporosis, dyslipidemia, gastrointestinal side effects, ocular side effects, immunosuppression and risk of infections, myopathy and cutaneous side effects [31].

The patient from our case report was also on long-term budesonide administered via a nebulizer. As presented by Christensson et al., the inhalation route of this glucocorticoid carries a low risk of systemic adverse effects, even in pregnant women and children [32]. Another drug that is used in the treatment of insulinoma in dogs and ferrets in combination with prednisone is diazoxide. It reduces insulin excretion, increases glycogenolysis and gluconeogenesis, and inhibits tissue uptake of glucose [1,3,4,30]. Importantly, it enables a reduction in the prednisone dose [2]. The dose of diazoxide recommended for ferrets is 5–10 mg/kg BID, and it can be gradually increased to 30 mg/kg BID [1,10]. Eventually, both drugs—prednisone and diazoxide—are no longer sufficient [2]. Diazoxide given in an initial dose in combination with prednisone gave no satisfactory effects. It can not be excluded that this dosing is inappropriate for rats.

Octreotide is a somatostatin analog that inhibits insulin secretion and is used in humans and dogs with insulin-secreting islet cell neoplasia [1,4,30]. In a study performed by Schmid and Brueggen, octreotide did not cause significant changes in plasma glucose levels in rats. However, the administration of another somatostatin analog—pasireotide—resulted in a quick glucose normalization in rats [33].

Besides dexamethasone, we also prescribe hepatoprotective drugs such as those containing SAMe in a dose of 20 mg/kg SID or phospholipids and ornithine. We also recommend diet modification—giving small, frequent meals while avoiding simple sugars [4]. As regards monitoring the patient, we suggest regular check-ups every 3–4 weeks, blood tests (especially with liver parameters) and performing glucose curves.

As could have been predicted, the patient’s glucose level dropped below the reference range with time despite an increase in the dexamethasone dosage. Based on the information provided by the owner, we assume that the sudden deterioration and death of the patient were associated with progressive hypoglycemia due to insulinoma.

## 4. Conclusions

Insulinoma should always be considered in cases of neurological deficits in rats, as it may cause peripheral neuropathy and paraparesis. Taking into account the insidious nature of these tumors in pet rats, in our opinion, it is worth measuring the glucose level in patients over 2 years of age on every visit, even when there are no obvious symptoms noted. It is highly recommended to perform abdominal ultrasound in every case of suspected functional pancreatic tumor. Dexamethasone should be considered as the drug of choice for the treatment of insulinoma in rats. It would also be good to advise clients on maintaining their pet rats in good body condition.

## Figures and Tables

**Figure 1 animals-12-02783-f001:**
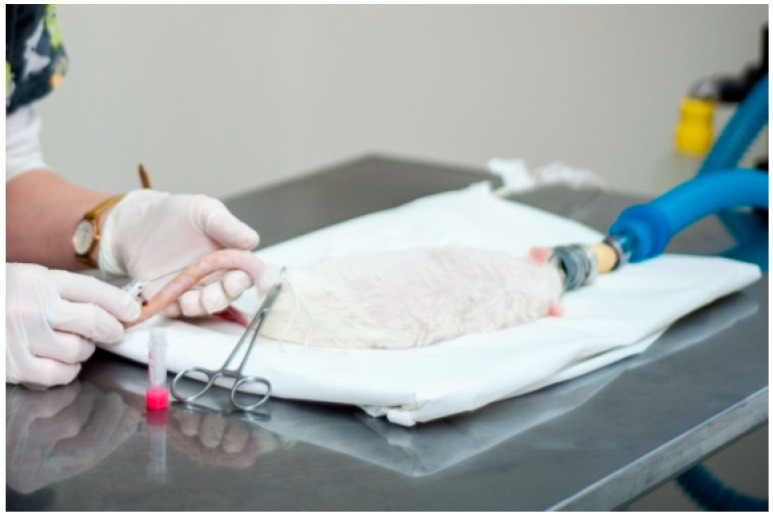
Procedure of blood collection from the lateral tail vein in the rat under isoflurane anesthesia.

**Figure 2 animals-12-02783-f002:**
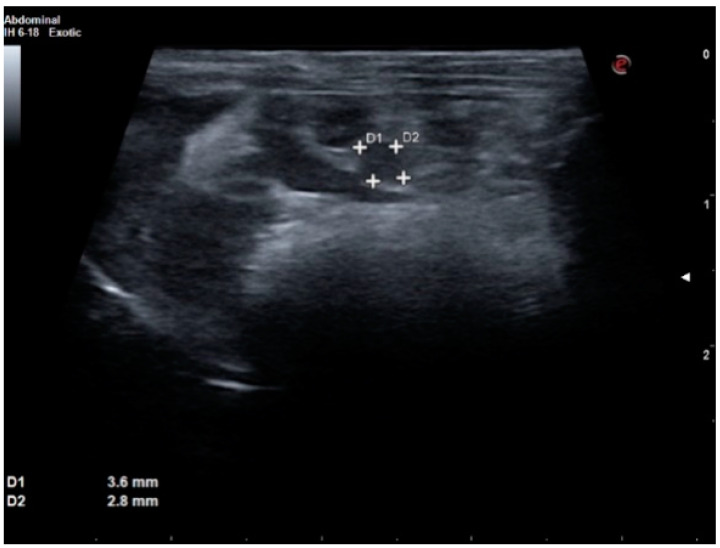
Ultrasound image of an insulinoma in the rat.

**Figure 3 animals-12-02783-f003:**
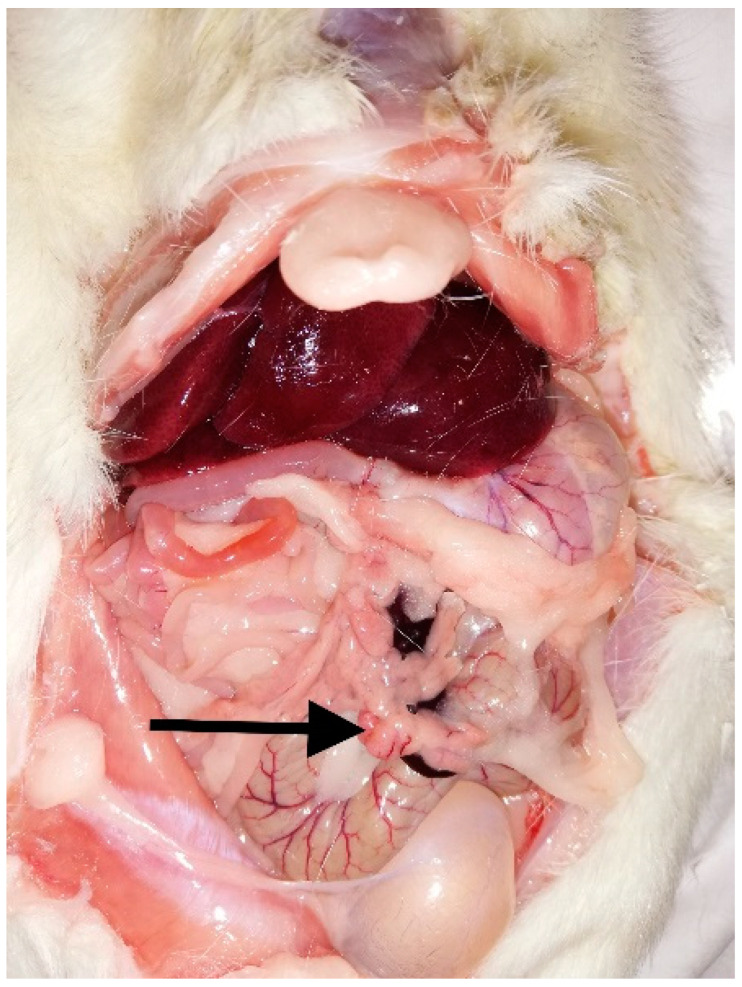
Postmortem image of the abdominal cavity of the rat. The arrow indicates a mass within the pancreas.

**Table 1 animals-12-02783-t001:** Blood results and reference ranges.

Parameter	Blood Results before Starting Treatment with Dexamethasone	Blood Results after 5 Weeks of Taking Dexamethasone	Reference Range According to Carpenter [10] and He et al. [11] *
RBC	8.35 T/l	8.42 T/l	7–8 T/l
Ht	43.2%	48.1%	35–45%
Hb	15.4 g/dL	16.5 g/dL	12–18 g/dL
MCV	51.7 fL	57.1 fL	55.21–64.8 fL *
MCH	18.4 pg	19.6 pg	18.7–21.2 pg *
MCHC	35.6 g/dL	34.3 g/dL	31.8–34.7 g/dL *
WBC	3.11 G/L	2.71 G/L	5–23 G/L
neutrophils	1.12 G/L	1.98 G/L	10–50%
lymphocytes	1.58 G/L	0.5 G/L	50–70%
monocytes	0.357 G/L	0.220 G/L	0–10%
eosinophils	0.03 G/L	0.003 G/L	0–5%
basophils	0.01 G/L	0.003 G/L	0–1%
PLT	647 G/L	465 G/L	840–1240 G/L
ALT (GPT)	32.0 U/L	77.4 U/L	20–92 U/L
AST (GOT)	135.6 U/L	186.6 U/L	
alpha-Amylase	534.2 U/L	488.9 U/L	
TP	65.1 g/L	73.7 g/L	56–76 g/L
CK	549 U/L	382.0 U/L	
GLDH	14.4 U/L	98.6 U/L	
creatinine	0.64 mg/dL	0.81 mg/dL	0.2–0.8 mg/dL
BUN	27.38 mg/dL	58.37 mg/dL	15–21 mg/dL
glucose	20.88 mg/dL	94.5 mg/dL	50–135 mg/dL
K^+^		5.57 mmol/L	5.9 mmol/L
Na^+^		146.5 mmol/L	135–155 mmol/L
Cl^-^		104.7 mmol/L	

The reference ranges are from Carpenter, only the ones with * are from He et al.

## Data Availability

The data presented in this study are available on request from the corresponding author.

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
