# Peer review of "New Insight on Insulinoma Treatment in a Pet Rat—A Case Report"

_animals, 2022, doi:10.3390/ani12202783_

Round 1

Reviewer 1 Report

xx1.     The authors should provide the figure of insulinoma by histopatholgical observation.

2.     The rat presented with respiratory syndrome at first visit which had been treated by drugs for 3 months. How about the condition of heart, respiratory system, adrenal glands as well as pituitary glands evidence of histopathology.

3.     Insulinoma is a kind of rare neuroendocrine tumor which had been reported in a patient with type 2 diabetes mellitus those who did discontinue treatment that uncontrolled high level of blood glucose might induce the function of islet cell at pancreas and further becoming functional adenoma. The author mentioned that the rat showed body weight loss slowly in its history. Whether the authors consider the possibility of insulinoma due to diabetes mellitus.

4.     At the results after dexamethasone treatment of Table 1 the authors should mention these results was detected by how many days after dexamethasone treatment. 

5.     In the section of case report, the authors should present those things concerning the information of check-up by month for example raising environment, food taking, body weight, body condition score, heart sound/lung sound by auscultation, parameters of physical examination,

6.     In the section of discussion, the authors have to discuss why the blood glucose goes down even under dexamethasone treatment, the effect of continue nebulization and the side effect of dexamethasone, that is, should focus on the change of physical examination, treatment protocol and the etiology of death by histopathology.

Author Response

  1. The authors should provide the figure of insulinoma by histopatholgical observation

Unfortunately till today we were unable to get the HP image of our insulinoma from the lab. Nevertheless from the beginning we wanted to focus more on the treatment.

  1. The rat presented with respiratory syndrome at first visit which had been treated by drugs for 3 months. How about the condition of heart, respiratory system, adrenal glands as well as pituitary glands evidence of histopathology.

I feel awkward to write it, but no other tissue than the pancreatic tumor and sciatic nerve was submitted to the lab for HP.

  1. Insulinoma is a kind of rare neuroendocrine tumor which had been reported in a patient with type 2 diabetes mellitus those who did discontinue treatment that uncontrolled high level of blood glucose might induce the function of islet cell at pancreas and further becoming functional adenoma. The author mentioned that the rat showed body weight loss slowly in its history. Whether the authors consider the possibility of insulinoma due to diabetes mellitus.

We didn't consider diabetes mellitus as the background for insulinoma. The rat was spayed when it had 6 months. The rat's weight generally fluctuated in the range of 321-368g with the highest score when it was around 1,5 y.o., being slightly overweight. After that time the weight gradually started to drop to 321g, when the rat was 2 y.o. We associated it with progressive CRD, older age and muscle loss. The rat didn't have any signs of diabetes mellitus. I have had only few rats with spontaneous diabetes. All of them were extremely obese and on a bad diet. I have had some rats with diabetes after local treatment with ointment or ocular drops containing dexamethasone. The rat from our report was neither that obese nor on a bad diet and it didn't get any other glucocorticoids than budesonid via inhalation route. I have never encountered diabetes after nebulisations in any of my patients. That is why I  didn't take diabetes into account and I didn't check it in this case. Previous blood exam was done long time before.

  1. At the results after dexamethasone treatment of Table 1 the authors should mention these results was detected by how many days after dexamethasone treatment. 

I have modified the text,  it is still in weeks, but I hope it is more precise now "After 5 weeks of taking dexamethasone, in a dosage of 1/36 tablet BID for the last 3 weeks..". And I have also put the information in the table 1.

  1. In the section of case report, the authors should present those things concerning the information of check-up by month for example raising environment, food taking, body weight, body condition score, heart sound/lung sound by auscultation, parameters of physical examination,

We came up with the assumption that we would write only about abnormalities found in clinical examination. Activity level and food taking was always the same and good. At least according to the owner. I have added some more information to the text, especially concerning weight and BCS. Do you want me to write about all parameters of physical examination during every check up?  I'm not quite sure what you mean by "raising environment".

  1. In the section of discussion, the authors have to discuss why the blood glucose goes down even under dexamethasone treatment, the effect of continue nebulization and the side effect of dexamethasone, that is, should focus on the change of physical examination, treatment protocol and the etiology of death by histopathology.

I added a paragraph to the discussion on why the blood glucose drops despite the GCs treatment, potential side effects of systemic treatment with GCs and safety of budesonide nebulisations. I also brought up the topic of death etiology in this case and deleted it from the case report part.

Reviewer 2 Report

General comments:

This manuscript is not in the typical format of the Animals journal, it makes it difficult to visualize, and because there are no number in the lines, it makes it difficult to point out the correction.

Specific comments:

Abstract

- First line: I suggest to start with  " This is a case report of an older rat...".

Introduction:

- The first paragraph does not have any bibliographic references, the authors should add some.

- Also I suggest to add in the abstract some of the informations mentioned here, such as: "insulinomas are insulin-producing tumors of pancreatic b-cells islets; and insulinomas have high prevalence in ferrets; and also spontaneous  insulinoaas in pet rats have been reported....

Case Report

- In regard to the anesthesia induction in a chamber, I suggest to add a photo. I think if the authors could add more photos to the case report it would be interesting for the readers. 

- When talking about " massaging of the muscles of the hind legs several times"", I suggest the exact number. How many times? 6 to 8 times a day? Did the authors also performed the flexor/withdrawl reflex ?

- when the authors talk about the seizures, why there was not a preventive planed pharmacological treatment in case this happened?  It would be expected as a negative outcome.

- In geral, when talking about the case report, I suggest to change the main structure of the presentation. The authors should start with past history and anamnesis, clinical signs, physical examination, complementary diagnostic exams , problems list , differentials, diagnostic, treatment and progression/outcome, finishing with the necropsy. 

Discussion

- Bibliographic references have to be added. There are paragraph that do not cite any reference. I suggest that the authors should improve the bibliography.

- Why does the hyperinsulinemia was associated with microangiopathic change? The authors should explain the pathophysiology in the discussion..

Author Response

This manuscript is not in the typical format of the Animals journal, it makes it difficult to visualize, and because there are no number in the lines, it makes it difficult to point out the correction.

I have switched on the counting of the lines.

Specific comments:

Abstract

- First line: I suggest to start with  " This is a case report of an older rat...".

I have changed this sentence according to your suggestion.

Introduction:

- The first paragraph does not have any bibliographic references, the authors should add some.

That's true.. I'm sorry for this oversight. But finally I have decided to delete this paragraph and put this information into the abstract, according to your suggestions below.

- Also I suggest to add in the abstract some of the informations mentioned here, such as: "insulinomas are insulin-producing tumors of pancreatic b-cells islets; and insulinomas have high prevalence in ferrets; and also spontaneous  insulinoaas in pet rats have been reported....

as above

Case Report

- In regard to the anesthesia induction in a chamber, I suggest to add a photo. I think if the authors could add more photos to the case report it would be interesting for the readers. 

I added a photo of blood sampling from the lateral tail vein under gas anesthesia. I also have a photo of a chest xray of this patient, but I'm not quite sure whether it  is relevant to the topic..

- When talking about " massaging of the muscles of the hind legs several times"", I suggest the exact number. How many times? 6 to 8 times a day? Did the authors also performed the flexor/withdrawl reflex ?

I have further specified the frequency of massages that was recommended.

Yes, we did perform this reflex and I have also added this information to the text.

- when the authors talk about the seizures, why there was not a preventive planed pharmacological treatment in case this happened?  It would be expected as a negative outcome.

I have developed this topic into: "According to the owner's report, the rat was presenting generalized muscular weakness, was anxious and breathing fast. The owner didn't recognize those symptoms as related to hypoglycemia, so she did not initiate hypoglycemic emergency treatment. When the rat started to have seizures the owner was already on her way to the clinic, but the rat died before getting  there."

By hypoglycemic treatment I meant honey and injection of dexamethasone.

- In geral, when talking about the case report, I suggest to change the main structure of the presentation. The authors should start with past history and anamnesis, clinical signs, physical examination, complementary diagnostic exams , problems list , differentials, diagnostic, treatment and progression/outcome, finishing with the necropsy. 

I don't quite understand, as in my opinion our case report has almost exactly the structure that you suggested. I have just added the differential diagnosis.

Discussion

- Bibliographic references have to be added. There are paragraph that do not cite any reference. I suggest that the authors should improve the bibliography.

I have checked that now every paragraph contains references besides abstract, conclusions and authors' opinions. Is it ok now or I'm still omitting something?

- Why does the hyperinsulinemia was associated with microangiopathic change? The authors should explain the pathophysiology in the discussion..

Findings of Sugimoto (2003) at al. " indicate that long-term hyperinsulinemic hypoglycemic insults may introduce generalized peripheral neuropathy complicated with endoneurial microangiopathy. our current findings suggest that hyperinsulinemia may play a pathogenic role in the development of endoneurial microangiopathy in diabetic patients, particularly those receiving excess insulin (...) But they didn't discuss the ethiology.

Eugene J Barrett at al. wrote "We have summarized both the general molecular processes involved in diabetic microvascular disease and many of their tissue-specific expressions. Clearly, there is much we still do not understand, and consequently, our ability to successfully intervene to prevent or reverse microvascular disease is quite limited." The article is from 2017.

Taking this into account I'm not able to explain the pathophysiology of this process.

Round 2

Reviewer 1 Report

I have no comments

Reviewer 2 Report

The authors cope with every suggestion made, and I think the manuscript was much improved after this revisions. I think it should be accepted for publication.